# The Cajal Body in Plant-Virus Interactions

**DOI:** 10.3390/v12020250

**Published:** 2020-02-23

**Authors:** Yi Ding, Rosa Lozano-Durán

**Affiliations:** 1Shanghai Center for Plant Stress Biology, Chinese Academy of Sciences, Shanghai 201602, China; yiding@psc.ac.cn; 2Center for Excellence in Molecular Plant Science, Chinese Academy of Sciences, Shanghai 201602, China; 3University of the Chinese Academy of Sciences, Beijing 100049, China

**Keywords:** Cajal body, plant-virus interactions, virus, RNPs, RdDM

## Abstract

Cajal bodies (CBs) are nuclear membraneless bodies composed of proteins and RNA. Although it is known that CBs play a role in RNA metabolism and the formation of functional ribonucleoprotein (RNP) particles, the whole breadth of CB functions is far from being fully elucidated. In this short review, we will summarize and discuss the growing body of evidence pointing to an involvement of this subnuclear compartment in plant-virus interactions.

## 1. Cajal Bodies in Plants

Cajal bodies (CBs; also known as accessory bodies or coiled bodies) are membraneless subnuclear organelles first described in 1903 by the cytologist Santiago Ramon y Cajal [1]. These nuclear bodies are present in all eukaryotes examined to date; their size (0.2–2 μm) and number (1–6 in animals; 1–2 in plants) vary depending on cell type, cell cycle status, and metabolic activity (reviewed in [2]) (Figure 1).

The defining component of CBs is a protein named coilin, considered the signature marker of this organelle. Strikingly, and despite the conservation of coilin across vertebrates and in plants, functional information about this protein is limited [3]; it has been demonstrated, however, that coilin is required for plant CB formation [4,5]. CBs are both physically and functionally associated with another subnuclear body, the nucleolus, with which they share components (e.g. fibrillarin) [6,7]. CBs are known to participate in the maturation of certain nuclear RNA species and in the formation of small nuclear ribonucleoproteins (snRNPs) involved in transcription, splicing, telomere maintenance, and ribosome biogenesis (reviewed in [6,8,9,10,11,12,13,14]). In plants, some elements of the RNA-directed DNA methylation (RdDM) pathway are concentrated in CBs [15,16], so this compartment has been proposed to be a site for the assembly of protein/RNA complexes mediating de novo DNA methylation [17,18]. In addition to the previously mentioned functions of CBs, these organelles also seem to influence cell cycle and DNA repair, growth, and developmental responses [2,11,19,20].

In recent years, multiple works have drawn a connection between plant CBs and responses to abiotic and biotic stresses (reviewed in [2]). Nevertheless, whether these functional effects are directly linked to RNA metabolism and/or snRNP formation or, on the contrary, rely on other functions of CBs remains to be clarified. Particularly intriguing is the interplay between CBs and the infection by viruses, which is emerging as a relevant aspect in multiple plant-virus interactions [5,21,22,23,24,25,26,27,28] (Table 1); the nature of this relationship, however, might vary depending on viral properties as discussed below.

## 2. The Emerging Role of the Cajal Body in Plant-Virus Interactions

In recent years, the CB has started to gain visibility in the plant virology field: multiple independent studies of the viral infection by both DNA and RNA viruses have unbiasedly converged on this subnuclear compartment. To begin with, a number of virus-encoded proteins have been found to interact with CB components and/or localize to CBs (Table 1; Figure 2). For example, the P2 protein from *Rice stripe virus* (RSV) (Fam. *Phenuiviridae*, genus *Tenuivirus*; (−)ssRNA genome), the nuclear inclusion protein A (NIa) from *Potato virus A* (PVA) (*Potyviridae*, *Potyvirus*; (+)ssRNA genome), and the protein encoded by open reading frame 3 (ORF3) from *Groundnut rosette virus* (GRV) (*Tombusviridae*, *Umbravirus*; (+)ssRNA genome) interact with the CB and nucleolar component fibrillarin [21,22,23,25]. The movement protein (TGBp1) from *Poa semilatent virus* (*Virgaviridae*, *Hordeovirus*; (+)ssRNA genome) and the 16K protein from *Tobacco rattle virus* (*Virgaviridae*, *Tobravirus*), on the other hand, interact with the signature CB protein coilin [24,26]. The V2 protein from *Tomato yellow leaf curl virus* (TYLCV) (*Geminiviridae*, *Begomovirus*) interacts with AGO4 mostly in CBs, where it accumulates [27]. The V2 protein from another geminivirus, *Grapevine red blotch-associated virus* (*Geminiviridae*, *Grablovirus*) can also be detected in CBs [29], but whether an interaction with AGO4 is conserved remains to be determined. Another example of a viral protein that can be found in CBs, but for which no CB-localized interactor has been found to date, is the p23 protein from *Citrus tristeza virus* (CTV) (*Closteroviridae*, *Closterovirus*) [21].

In some instances, a specific function of CBs (or the nucleolus) in the viral cycle has either been demonstrated or can be hypothesized. For some RNA viruses, the association with CBs/nucleolus is essential for systemic infection; that is the case for the umbravirus GRV, which does not encode a canonical capsid protein (CP). In the absence of a CP, it is the virus-encoded ORF3 protein that associates with the viral genome in order to protect it, forming filamentous ribonucleoproteins (RNPs) that move long-distance in the plant [30,31]. During GRV infection, the ORF3 protein has a nuclear phase, passing through CBs and the nucleolus before returning to the cytosol, a cycling that is essential for the formation of viral RNPs [25,32,33,34]. It has been shown that the ORF3 protein interacts with fibrillarin in CBs, triggering a reorganization of this subnuclear structure into multiple CB-like bodies that ultimately fuse with the nucleolus [34]. Interestingly, fusion of the ORF3 protein-induced CB-like structures with the nucleolus (Figure 2), which requires the interaction between the ORF3 protein and fibrillarin, is essential for the viral protein to reach this compartment, and therefore for RNP formation and systemic viral infection [25,34]. Notably, fibrillarin is comprised in cytoplasmic viral RNPs [25], indicating that this plant protein is sequestered by the ORF3 protein at its passage through the nucleus. Taken together, these results suggest that GRV has evolved to hijack intracellular trafficking pathways involving CBs, fibrillarin, and the nucleolus in order to assemble transport-competent viral RNPs and allow for the systemic invasion of the plant.

Other proteins encoded by RNA viruses, e.g. P2 from RSV and TGB1 from *Barley stripe mosaic virus*, might promote viral movement through a strategy similar to that used by the ORF3 protein, involving fibrillarin and the nucleolus [23,35]. The P23 protein of CTV shows CB and nucleolar localization, but it is also targeted to plasmodesmata, which would be in line with a function in intercellular and potentially long-distance movement [21]. P23, however, has also been described to act as a viral suppressor of RNA silencing, a function for which its nucleolar localization might be required [21]; whether CB localization plays a role in facilitating silencing suppression remains to be determined. The NIa protein of PVA, which also interacts with fibrillarin, acts as a silencing suppressor in a CB/nucleolus localization-dependent manner; however, depletion of fibrillarin does not impact long-distance viral movement [22], suggesting that the viral subversion of CBs/nucleolus can serve independent purposes. This idea is further supported by the fact that NIa interacts with fibrillarin in the nucleus, but not in the cytoplasm [22]. Of note, coilin-silenced *N. benthamiana* plants, in which no CBs are detected, retain post-transcriptional gene silencing (PTGS) activity in local transient assays [5]. It needs to be considered, nevertheless, that silencing does not result in a complete loss of coilin, which could therefore partially retain its function(s).

A different scenario is that of the infection by DNA viruses. The genomes of DNA viruses are targeted by DNA methylation, which is considered an effective anti-viral defense [36]. In plants, the establishment of de novo DNA methylation is regulated by the RNA-directed DNA methylation (RdDM) pathway. The canonical RdDM pathway involves two RNA polymerase II-related proteins, Pol IV and Pol V. Pol IV produces RNA transcripts that are subsequently converted to dsRNA by RDR2, then cut into 24-nt small interfering RNA (siRNA) by DCL3. These siRNA are then loaded into AGO4, which finds scaffold RNA molecules synthesized by Pol V and recruits the methyl transferase DRM2 [37]. Ultimately, DRM2 methylates adjacent DNA sequences, creating a chromatin environment unfavorable to transcription, and hence leading to transcriptional gene silencing (TGS). Intriguingly, the core RdDM component AGO4 accumulates in CBs, while it is not present in the nucleolus [17], and a coilin mutant in *Arabidopsis thaliana* displays lower levels of AGO4 [18]. Other central players in RdDM, namely the Pol V subunit NRPE1, DCL3, and RDR2, at least partially co-localize with AGO4 in CBs [17,18,38,39]. Considering these results, it has been proposed that the CB is a center for the assembly of AGO4/Pol V/siRNA complexes participating in RdDM [17]; whether coilin-silenced CB-depleted plants display a functional RdDM is at this point unknown.

In order to achieve a successful infection, DNA viruses have evolved TGS suppressors. This is the case of V2 from TYLCV which, among other functions, can interact with AGO4, suppressing its binding to and subsequent methylation of the viral genome [27,40]. The interaction between V2 and AGO4 takes place mainly in CBs, and yet is sufficient to suppress methylation of the entire viral population, suggesting a connection between CBs and viral DNA replication [27]. Remarkably, CB localization is required for the TGS suppression function, since impairment of the former abolished the latter; in agreement with this, knock-down of coilin by virus-induced gene silencing (VIGS) dramatically reduced methylation of the viral DNA [27]. Taken together, these results draw a functional connection between CBs, RdDM, and anti-viral DNA methylation. The function of V2 as a suppressor of viral DNA methylation in CBs might be conserved across geminiviruses, since the V2 protein encoded by other geminivirus species also localizes to this subnuclear compartment or, at least, to CB-like structures [29,41], and V2 from the geminivirus *Cotton leaf Multan virus* can also interact with AGO4 [42].

The functional relevance of the CB (or, more specifically, of its signature component coilin) has been assessed in an unbiased manner using transgenic *N. benthamiana* and tobacco plants in which coilin has been silenced [5,24]. In these plants, CBs cannot be detected, and are therefore a perfect model system to investigate the role of this subnuclear compartment. Interestingly, coilin silencing had an effect on the infection by all tested viruses, including RNA and DNA viruses, although a general rule is not easy to infer [5]. Following inoculation with *Tobacco rattle virus* (TRV) or *Tomato black ring virus* (*Secoviridae*, *Nepovirus*), coilin-silenced plants do not show recovery from the infection, as opposed to wild-type plants; along the same lines, infection by *Barley stripe mosaic virus* (BSMV) (*Virgaviridae*, *Hordeovirus*) and *Tomato golden mosaic virus* (*Geminiviridae*, *Begomovirus*) results in the development of stronger symptoms in coilin knocked-down plants. These results suggest that CBs/coilin play a crucial role in anti-viral defense against these viral species. The exact underlying mechanism, however, is unclear: coilin silencing does not affect neither TRV nor BSMV replication in protoplasts; and while for TRV differences in viral accumulation are found in the young tissues where recovery takes place, the increased BSMV accumulation can be detected locally and systemically at early stages of the infection only [5]. In contrast with the previous results, coilin-silenced plants show milder symptoms upon infection by *Potato virus Y* (PVY) (*Potyviridae*, *Potyvirus*) and *Turnip vein clearing virus* (*Virgaviridae*, *Tobamovirus*), pointing to a positive role of coilin in the viral cycle. In the case of PVY, such a role might be related to viral replication, since knock-down of coilin leads to reduced viral accumulation both in protoplasts and inoculated leaves, although not in systemic leaves [5].

Subsequent investigations shed light on the effect of coilin on the infection by TRV [24]: the TRV-encoded silencing suppressor 16K physically associates with coilin and triggers its re-localization from CBs to the nucleolus, a change that is potentially perceived by the plant, since it correlates with the activation of salicylic acid (SA) accumulation and signaling. In turn, SA responses restrict TRV invasion, allowing for the recovery of the infected plant. This coilin- and SA-dependent defense mechanism may act in parallel to gene silencing to ensure restriction of the invading virus. The molecular link between the coilin/16K interaction and the activation of SA-dependent defenses awaits to be uncovered.

## 3. Conclusions and Outlook

A growing body of work is making the CB emerge as a relevant player in plant-virus interactions, based on localization and interactions of adapted virus-encoded proteins, as well as on functional genetic data. However, considering the results currently available in the literature, general patterns have not so far become evident; the nature of the involvement of CBs in the infection by a particular virus might depend on biological properties of the species, hence its requirements and evolved strategies for replication and spread. Although an involvement of CBs in the assembly of viral RNPs, and hence systemic movement, has been clearly demonstrated for the RNA virus GRV, and a link between CBs and RdDM-mediated methylation of the viral genome seems likely for the DNA virus TYLCV, CBs may exert a plethora of other functions during the viral subversion of the host cell. A broad and comprehensive analysis of the role of CBs across diverse plant-virus interactions will likely be required for commonalities to arise.

## Figures and Tables

**Figure 1 viruses-12-00250-f001:**
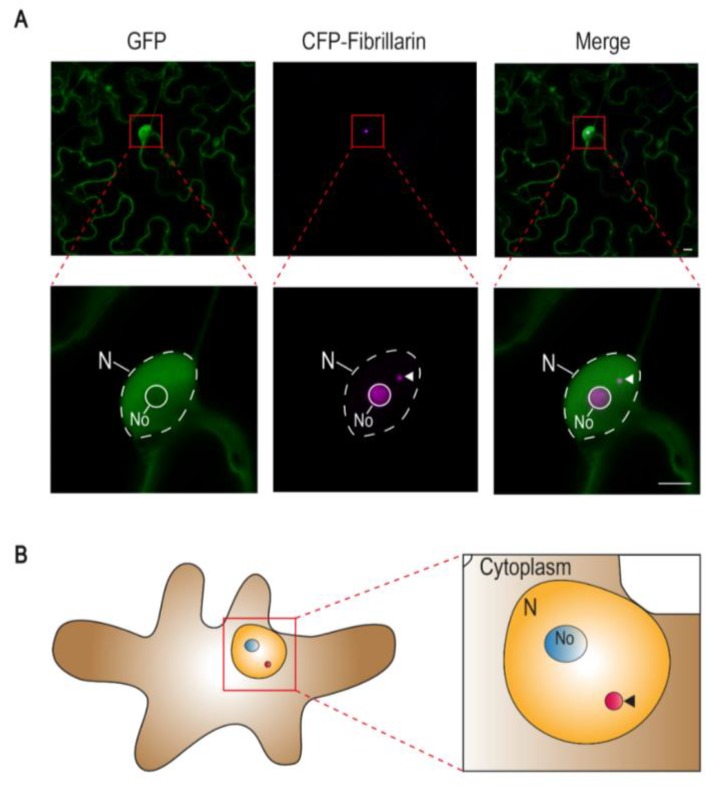
The Cajal body in plants. (**A**) Confocal micrographs of *Nicotiana benthamiana* epidermal pavement cells transiently expressing free green fluorescent protein (GFP) and the nucleolus and Cajal body marker fibrillarin fused to the cyan fluorescent protein (CFP-fibrillarin). Scale bar: 5 μm. (**B**) Schematic representation of a *N. benthamiana* epidermal pavement cell showing the nucleus, the nucleolus, and a Cajal body. N: Nucleus; No: Nucleolus. In both (**A**) and (**B**), arrowheads indicate the Cajal body.

**Figure 2 viruses-12-00250-f002:**
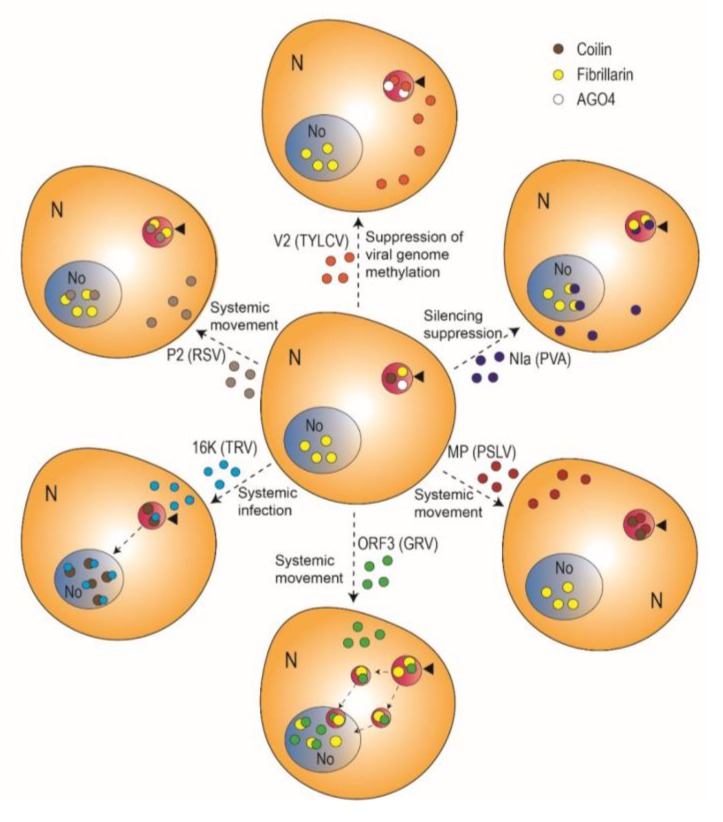
Plant virus-encoded proteins described as localized in Cajal bodies and/or interacting with Cajal body components. Arrowheads indicate Cajal bodies; dotted arrows denote movement. See text for details. TYLCV: *Tomato yellow leaf curl virus*; PVA: *Potato virus A*; PSLV: *Poa semilatent virus*; GRV: *Groundnut rosette virus*; TRV: *Tobacco rattle virus*; RSV: *Rice stripe virus*.

**Table 1 viruses-12-00250-t001:** Plant virus-encoded proteins described as localized in the Cajal body and/or interacting with Cajal body components.

Virus Name	Family	Genus	Genome	Virus-Encoded Protein	Localization	Known Cajal Body Interactor	Described Function	Ref.
*Rice stripe virus* (RSV)	*Phenuiviridae*	*Tenuivirus*	(−)ssRNA	P2	Nucleus, cytoplasm, Cajal bodies, nucleolus	Fibrillarin	Systemic movement	[23]
*Potato virus A* (PVA)	*Potyviridae*	*Potyvirus*	(+)ssRNA	Nuclear inclusion protein a (NIa)	Nucleus, nucleolus, Cajal bodies	Fibrillarin	Suppression of RNA silencing	[22]
*Groundnut rosette virus* (GRV)	*Tombusviridae*	*Umbravirus*	(+)ssRNA	Open reading frame 3 (ORF3) protein	Cajal bodies, nucleolus, Cajal body (CB)-like structures	Fibrillarin	Systemic movement	[25]
*Poa semilatent virus* (PSLV)	*Virgaviridae*	*Hordeovirus*	(+)ssRNA	Movement protein (MP; TGBp1)	Nucleolus, Cajal bodies, inclusions in the nucleoplasm	Coilin	Cell-to-cell and systemic movement	[26]
*Tobacco rattle virus* (TRV)	*Virgaviridae*	*Tobravirus*	(−)ssRNA	16K	Nucleus, nucleolus, Cajal bodies	Coilin	Systemic infection	[24]
*Tomato yellow leaf curl virus* (TYLCV)	*Geminiviridae*	*Begomovirus*	ssDNA	V2	Cajal bodies, nucleoplasm, cytoplasm	AGO4	Suppression of viral DNA methylation	[27]
*Grapevine red blotch-associated virus* (GRBaV)	*Geminiviridae*	*Grablovirus*	ssDNA	V2	Nucleoplasm, Cajal bodies, cytoplasm, nucleolus	Fibrillarin	Unknown	[29]
*Citrus tristeza virus* (CTV)	*Closteroviridae*	*Closterovirus*	(+)ssRNA	P23	Nucleolus, Cajal bodies, plasmodesma	Unknown	Suppression of RNA silencing	[21]

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
