# Peer review of "The Cajal Body in Plant-Virus Interactions"

_viruses, 2020, doi:10.3390/v12020250_

Round 1

Reviewer 1 Report

The intranuclear Cajal bodies of eukaryotes have been extensively investigated, since they serve primarily as assembly compartments for small nuclear RNPs (snRNPs) (Ohtani, 2018) and serve as hub for the regulated splicing of various mRNAs (secondary effects). Correspondingly, a multitude of changes in gene regulation in various fields have been described. For viruses, CBs may have positive and negative effects. Whereas animal viruses, such as Adenovirus and Herpesvirus, have been investigate in close detail, the role of CBs for plant viruses have emerged in the last two decades only. A comprehensive review with a focus on plant viruses has recently appeared (Xu et al., 2020). The authors of the submitted manuscript update the same issue and provide some complementary discussions, which are worth to be published. However, the text has a great potential for improvement concerning major topics and details of information.

Throughout the text, a clearer discrimination between the primary and secondary effects of CBs would be helpful, as well as between pro-viral and anti-viral effects. Some examples for animal viruses should be included. A more careful description of knockdown consequences is necessary. The term „interact“ should be used only, if direct physical interaction has been proven; otherwise (co-)localize in/to or other more precise descriptions are appropriate.

Minor comments and suggestions:

31 „functional information about this protein is lacking“; too strong: see (Machyna et al., 2015)

33-36 “CBs are known to play a role in RNA metabolism, particularly in the maturation of certain nuclear RNA species, and in the formation of ribonucleoproteins (RNPs) involved in transcription, splicing, telomere maintenance, and ribosome biogenesis (reviewed in [5,7-12])”; close to plagiarism; re-write and provide the primary literature sources for the different topics. “RNA metabolism” may be a misleading term here: primary they are assembly platforms for snRNPs / “snRNP biogenesis?”(Ohtani, 2018)

37-39 “In plants, some elements of the RNA-directed DNA methylation (RdDM) pathway are concentrated in CBs [13,14], so this compartment has been proposed to be the site of assembly of protein/RNA complexes mediating this process [15,16]”; In plants, elements of the RNA-directed DNA methylation (RdDM) pathway may be concentrated in CBs [13,14] to mediate the assembly of protein/RNA complexes of this process [15,16] ?

39-41 “In addition to the previously mentioned and considered canonical functions of CBs, these organelles also seems to influence cell cycle and DNA repair, growth, and developmental responses [2,10,17,18].” re-write. What are the canonical functions? specify the primary references for the different topics, not just review articles. – or use “reviewed in...”

42-45 “In recent years, multiple works have drawn a connection between plant CBs and responses to abiotic and biotic stresses (reviewed in [2]), although whether these functional effects are directly linked to RNA metabolism and/or RNP formation or on the contrary rely on other, yet unidentified function of CBs remains to be clarified”; weak sentence, re-write and split informations

45-48 “Particularly intriguing is the interplay between the Cajal body and the infection by viruses, which is emerging as a relevant aspect in multiple plant-virus interactions [4,19-25] (Table 1); the nature of this relationship, however, might vary depending on viral properties, as discussed below.”; why “Particularly intriguing”? split sentence. sort animal and plant viruses; proviral and antiviral interactions? add primary references for animal viruses and recent review (see (Xu et al., 2020))

Table 1: Consider the geminivirus ICMV AV2 protein for the dynamics of the localization (Rothenstein et al., 2007)?

52-54 Weak and complicated sentence, re-write

56 “Fam. Phenuivirudae”; replace Phenuiviridae. Here and further one “Fam.” and “genus” can be deleted in order to facilitate reading

59-60 “all interact with the CB and nucleolar component fibrillarin”; interact with the CB and nucleolar component fibrillarin

61 “Hoerdeovirus”; Hordeovirus

69 nucleolar localization signal has been defined!

64 “interacts with AGO4 mostly in the CB, where it accumulates”; mostly ?; rewrite. see also (Rothenstein et al., 2007)

76-80 “For some RNA viruses, the association with the CB/nucleolus is essential for systemic infection; that is the case for GRV which, like all umbraviruses, does not encode a canonical capsid protein (CP). In the absence of a CP, it is the virus-encoded ORF3 that associates to the viral genome in order to protect it, forming filamentous RNPs that move long distance in the plant [28,29].”; For some RNA viruses, the association with the CB/nucleolus is essential for systemic infection: GRV, an umbravirus, does not encode a capsid protein (CP), but the virus-encoded ORF3 protein associates to the viral genome and forms filamentous RNPs that move long distance in the plant [28,29]. use ORF 3 protein furtheron, since an ORF cannot interact!

87 “components of the viral RNPs in the cytoplasm [8],”; check reference; rewrite sentence

92-94 comparison to ORF3 protein is not relevant here.

98 split sentence

99-102 rewrite and split

103 reference?

103 -105 new paragraph and more detailed discussion: Knockdown of coilin is not complete, absence of CBs does not exclude function of coilin in smaller aggregates.

118-119 “that the CB is the center of assembly of AGO4/Pol V/siRNA”; that the CB is a center for the assembly of AGO4/Pol V/siRNA

121 V2 can also interact with SGS3 and influence PTGS (Glick et al., 2008)

123 “AGO4 takes place mainly in the CB”; mainly? This is not really checked in the reference.

123-126 poor sentence. Split and rewrite more precisely. So far, there is no co-localization between CB and replicating viral DNA. “viral DNA metabolism” is not the best term here. “required” is too strong.

129 “devoid of CBs”; knockdown was not complete

136 “coilin-deficient”, coilin-reduced

137 – 141 split sentence. “coiling silencing”; coilin silencing. Discriminate between pro- and anti-viral effects.

141-144 “In contrast with the previous results, coilin-silenced plants show milder symptoms upon infection by Potato virus Y (PVY) (Fam. Potyviridae, genus Potyvirus) and Turnip vein clearing virus (Fam. Potyviridae, genus Tobamovirus), pointing to a role of coilin in the viral cycle”; Coilin-silenced plants showed milder symptoms upon infection by Potato virus Y (PVY) (Potyviridae, Potyvirus) and Turnip vein clearing virus (Potyviridae, Tobamovirus), pointing to a role of coilin in the viral cycle

145-146 “has a negative impact on viral accumulation both in protoplasts and inoculated leaves”; reduced viral accumulation both in protoplasts and inoculated leaves, but not in systemically infected leaves.

147-154 The interesting point here is that 16k redirects coilin from CB to the nucleolus!

Ref.

Glick, E., Zrachya, A., Levy, Y., Mett, A., Gidoni, D., Belausov, E., Citovsky, V., Gafni, Y., 2008. Interaction with host SGS3 is required for suppression of RNA silencing by tomato yellow leaf curl virus V2 protein. Proc. Natl. Acad. Sci. USA 105, 157-161.

Machyna, M., Neugebauer, K.M., Stanek, D., 2015. Coilin: The first 25 years. RNA Biol 12, 590-596.

Ohtani, M., 2018. Plant snRNP Biogenesis: A Perspective from the Nucleolus and Cajal Bodies. Frontiers in Plant Science 8.

Rothenstein, D., Krenz, B., Selchow, O., Jeske, H., 2007. Tissue and cell tropism of Indian cassava mosaic virus (ICMV) and its AV2 (precoat) gene product. Virology 359, 137-145.

Xu, M., Mazur, M.J., Tao, X., Kormelink, R., 2020. Cellular RNA Hubs: Friends and Foes of Plant Viruses. Mol Plant Microbe Interact 33, 40-54.

Author Response

Response to Reviewer 1

The intranuclear Cajal bodies of eukaryotes have been extensively investigated, since they serve primarily as assembly compartments for small nuclear RNPs (snRNPs) (Ohtani, 2018) and serve as hub for the regulated splicing of various mRNAs (secondary effects). Correspondingly, a multitude of changes in gene regulation in various fields have been described. For viruses, CBs may have positive and negative effects. Whereas animal viruses, such as Adenovirus and Herpesvirus, have been investigate in close detail, the role of CBs for plant viruses have emerged in the last two decades only. A comprehensive review with a focus on plant viruses has recently appeared (Xu et al., 2020). The authors of the submitted manuscript update the same issue and provide some complementary discussions, which are worth to be published. However, the text has a great potential for improvement concerning major topics and details of information.

Throughout the text, a clearer discrimination between the primary and secondary effects of CBs would be helpful, as well as between pro-viral and anti-viral effects. Some examples for animal viruses should be included. A more careful description of knockdown consequences is necessary. The term „interact“ should be used only, if direct physical interaction has been proven; otherwise (co-)localize in/to or other more precise descriptions are appropriate.

  • We thank this reviewer for his/her careful review of our manuscript and his/her suggestions. Following the reviewer’s advice, we have modified the text to improve clarity and readability and have added relevant references (including Ohtani, 2018, and the recent review by Xu et al. (2020)).

We believe the distinction between “primary” and “secondary” functions of CBs is blurred and a bit artificial, limited by our current understanding, which is scarce. In most of the examples in which the CB seems to play a role during the viral infection, it is hard to ascribe a particular known function to the observed effect – this is highlighted in the text. In those cases in which a pro- or anti-viral effect of silencing coiling/depleting CBs has been observed, this is now clearly stated.

Although a thorough comparison between animal and plant viruses in connection with the CB is definitely of interest, we believe this is out of the scope of the current short review, which focuses on the emerging involvement of CB in plant-virus interactions, trying to provide an overview of the effects reported so far. Additionally, several observations hint at fundamental differences in CBs and their role in the viral infection between animals and plants: for example, RdDM, which seems linked to CBs, is plant-specific; viral movement, which in some cases seems to require CB components, occurs in very different manners in animals and plants; mutations in the core CB component coilin can have very different effects in plants and animals.

Following the reviewer’s indication, we have made sure that the term “interaction”, when referred to proteins, is only used when direct physical association has been proven.

Minor comments and suggestions:

31 „functional information about this protein is lacking“; too strong: see (Machyna et al., 2015)

  • This statement has now been modified to read “…is limited”; the reference to Machyna et al., 2015 has been added at the end of this sentence. Please note a similar statement in the recent review by Xu et al. (2020), line 712: “…its function [of coilin] remains elusive (Machyna et al., 2015)”.

33-36 “CBs are known to play a role in RNA metabolism, particularly in the maturation of certain nuclear RNA species, and in the formation of ribonucleoproteins (RNPs) involved in transcription, splicing, telomere maintenance, and ribosome biogenesis (reviewed in [5,7-12])”; close to plagiarism; re-write and provide the primary literature sources for the different topics. “RNA metabolism” may be a misleading term here: primary they are assembly platforms for snRNPs / “snRNP biogenesis?”(Ohtani, 2018)

  • This sentence has now been re-written, and the reference to Ohtani 2018 has been added; we have also removed the expression “RNA metabolism”. We have refrained from adding the primary literature sources, since we believe this can be considered basic knowledge, is contained in multiple excellent recent reviews, and is not plant-specific.

37-39 “In plants, some elements of the RNA-directed DNA methylation (RdDM) pathway are concentrated in CBs [13,14], so this compartment has been proposed to be the site of assembly of protein/RNA complexes mediating this process [15,16]”; In plants, elements of the RNA-directed DNA methylation (RdDM) pathway may be concentrated in CBs [13,14] to mediate the assembly of protein/RNA complexes of this process [15,16] ?

  • The fact that some proteins involved in RdDM (e.g. AGO4) are concentrated in CBs has long been known (see Li et al., 2006); however, the biological significance of this observation remains elusive. For that reason, we phrased the sentence carefully so as to not mislead readers.

39-41 “In addition to the previously mentioned and considered canonical functions of CBs, these organelles also seems to influence cell cycle and DNA repair, growth, and developmental responses [2,10,17,18].” re-write. What are the canonical functions? specify the primary references for the different topics, not just review articles. – or use “reviewed in...”

  • The term “canonical” has been removed, to avoid confusion; “reviewed in” has been added.

42-45 “In recent years, multiple works have drawn a connection between plant CBs and responses to abiotic and biotic stresses (reviewed in [2]), although whether these functional effects are directly linked to RNA metabolism and/or RNP formation or on the contrary rely on other, yet unidentified function of CBs remains to be clarified”; weak sentence, re-write and split informations

  • Following the reviewer’s advice, we have re-written this sentence and split it in two.

45-48 “Particularly intriguing is the interplay between the Cajal body and the infection by viruses, which is emerging as a relevant aspect in multiple plant-virus interactions [4,19-25] (Table 1); the nature of this relationship, however, might vary depending on viral properties, as discussed below.”; why “Particularly intriguing”? split sentence. sort animal and plant viruses; proviral and antiviral interactions? add primary references for animal viruses and recent review (see (Xu et al., 2020))

  • We consider this interplay particularly intriguing because independently evolved viruses with different cycles seem to have a functional connection with the CB, although common patterns do not arise clearly at the moment. We focus on plants specifically – this is indicated in the sentence (“plant-virus interactions”). The recent review by Xu et al. (2020) has been added.

Table 1: Consider the geminivirus ICMV AV2 protein for the dynamics of the localization (Rothenstein et al., 2007)?

  • We had not initially included this reference since, although AV2 can be observed in a nuclear body, whether this nuclear body is indeed the Cajal body has not been demonstrated. We have now added this reference to the text, to highlight, together with Guo et al., 2015 and Wang et al., 2019, that V2 from geminiviruses might have a conserved function in the CB.

52-54 Weak and complicated sentence, re-write

  • Following the reviewer’s advice, this sentence has been re-phrased.

56 “Fam. Phenuivirudae”; replace Phenuiviridae. Here and further one “Fam.” and “genus” can be deleted in order to facilitate reading

  • We thank the reviewer for spotting this mistake. Following his/her suggestion, “Fam.” And “genus” have been removed after the first mention.

59-60 “all interact with the CB and nucleolar component fibrillarin”; interact with the CB and nucleolar component fibrillarin

  • This sentence has been modified as per the reviewer’s suggestion.

61 “Hoerdeovirus”; Hordeovirus

  • We thank the reviewer for spotting this mistake, which has now been corrected.

69 nucleolar localization signal has been defined!

  • We apologize, but do not understand what the reviewer is referring to here.

64 “interacts with AGO4 mostly in the CB, where it accumulates”; mostly ?; rewrite. see also (Rothenstein et al., 2007)

  • V2 interacts with AGO4 mostly in the CB, but a faint signal can be detected in the nucleoplasm in BiFC assays; a fusion GFP-V2 protein, which does not localize in CBs, still interacts with AGO4 in BiFC (weakly) and co-IP assays – see Wang et al., 2019, bioRxiv (updated version).

76-80 “For some RNA viruses, the association with the CB/nucleolus is essential for systemic infection; that is the case for GRV which, like all umbraviruses, does not encode a canonical capsid protein (CP). In the absence of a CP, it is the virus-encoded ORF3 that associates to the viral genome in order to protect it, forming filamentous RNPs that move long distance in the plant [28,29].”; For some RNA viruses, the association with the CB/nucleolus is essential for systemic infection: GRV, an umbravirus, does not encode a capsid protein (CP), but the virus-encoded ORF3 protein associates to the viral genome and forms filamentous RNPs that move long distance in the plant [28,29]. use ORF 3 protein furtheron, since an ORF cannot interact!

  • We have modified this sentence, following the reviewer’s advice. We have also added “protein” to “ORF3” when referring to the encoded protein, in order to avoid ambiguity.

87 “components of the viral RNPs in the cytoplasm [8],”; check reference; rewrite sentence

  • We thank the reviewer for spotting this mistake; the reference has now been corrected. Following the reviewer’s advice, the sentence has also been re-written.

92-94 comparison to ORF3 protein is not relevant here.

  • Our intention here was to highlight the commonalities in localization first, then indicate the differences; nevertheless, following the reviewer’s advice, we have now removed the reference to the ORF3 protein.

98 split sentence

  • We apologize to the reviewer but we do not understand what he/she means here.

99-102 rewrite and split

  • This sentence has now been re-written and split in two.

103 reference?

  • The corresponding reference has now been added at the end of the sentence.

103 -105 new paragraph and more detailed discussion: Knockdown of coilin is not complete, absence of CBs does not exclude function of coilin in smaller aggregates.

  • We have now added a sentence to clarify this.

118-119 “that the CB is the center of assembly of AGO4/Pol V/siRNA”; that the CB is a center for the assembly of AGO4/Pol V/siRNA

  • This sentence has been modified following the reviewer’s suggestion.

121 V2 can also interact with SGS3 and influence PTGS (Glick et al., 2008)

  • The interaction between V2 and SGS3 occurs outside of the nucleus, consistent with the localization of SGS3. In order to not neglect that V2 is multifunctional, we have added “among other functions” at the relevant point in the text.

123 “AGO4 takes place mainly in the CB”; mainly? This is not really checked in the reference.

  • V2 interacts with AGO4 mostly in the CB, but a faint signal can be detected in the nucleoplasm in BiFC assays; in the version 2 of the cited preprint it is shown that a fusion GFP-V2 protein, which does not localize in CBs, still interacts with AGO4 in BiFC (weakly) and co-IP assays.

123-126 poor sentence. Split and rewrite more precisely. So far, there is no co-localization between CB and replicating viral DNA. “viral DNA metabolism” is not the best term here. “required” is too strong.

  • Following the reviewer’s advice, we have re-written this sentence, and modified “viral DNA metabolism” to “viral DNA replication”. We are aware that so far no co-localization between CB and replicating viral DNA has been demonstrated, and that a potential connection could be indirect – hence we have phrased this idea cautiously.

129 “devoid of CBs”; knockdown was not complete

  • We have highlighted in the text the fact that knockdown of coilin is not complete; however, CBs cannot be detected in the cells, which is why we mentioned they are devoid of CBs (albeit not of coilin). However, as indicated by the reviewer, the existence of smaller aggregates, or even smaller CBs not detectable by confocal microscopy, cannot be completely ruled out, so in order not to be misleading we have re-phrased this sentence to indicate that CBs cannot be detected.

136 “coilin-deficient”, coilin-reduced

  • We have modified this sentence following the reviewer’s suggestion.

137 – 141 split sentence. “coiling silencing”; coilin silencing. Discriminate between pro- and anti-viral effects.

  • We thank the reviewer for spotting this mistake, which has now been corrected. The indicated sentence has been split.

141-144 “In contrast with the previous results, coilin-silenced plants show milder symptoms upon infection by Potato virus Y (PVY) (Fam. Potyviridae, genus Potyvirus) and Turnip vein clearing virus (Fam. Potyviridae, genus Tobamovirus), pointing to a role of coilin in the viral cycle”; Coilin-silenced plants showed milder symptoms upon infection by Potato virus Y (PVY) (Potyviridae, Potyvirus) and Turnip vein clearing virus (Potyviridae, Tobamovirus), pointing to a role of coilin in the viral cycle

  • This sentence has now been modified as indicated by the reviewer.

145-146 “has a negative impact on viral accumulation both in protoplasts and inoculated leaves”; reduced viral accumulation both in protoplasts and inoculated leaves, but not in systemically infected leaves.

  • This sentence has now been modified following the reviewer’s suggestion.

147-154 The interesting point here is that 16k redirects coilin from CB to the nucleolus!

  • We agree with the reviewer in that this is a very interesting observation indeed. This was already mentioned in the previous version of the sentence, and has now been highlighted.

Reviewer 2 Report

This manuscript is a small review regarding Cajal bodies in plants, especially focused on plant viruses involvement. The manuscript is well written. I only suggest one issue: in literature a recent major review on Cajal bodies in plants  (Love et al 2017, cited by authors) also discussing their role in disease response, is present. As a consequence, only a small number of papers contribute to the novelty of this new review. In my opinion, anyway, this review is worth publishing in Viruses, because of its special focusing on plant viruses.

Author Response

Response to Reviewer 2

This manuscript is a small review regarding Cajal bodies in plants, especially focused on plant viruses involvement. The manuscript is well written. I only suggest one issue: in literature a recent major review on Cajal bodies in plants  (Love et al 2017, cited by authors) also discussing their role in disease response, is present. As a consequence, only a small number of papers contribute to the novelty of this new review. In my opinion, anyway, this review is worth publishing in Viruses, because of its special focusing on plant viruses.

  • We thank the reviewer for his/her positive assessment. We believe that the involvement of the CB in plant-virus interactions is an emerging topic, and therefore considered worthwhile gathering all the available information showing molecular or functional connections between this subnuclear compartment and plant viruses, including the most recent literature. Since recent excellent reviews have already covered the cellular functions of the CB as well as its role in stress responses in plants, we have opted for a short and concise format, centered on the interplay between this organelle and the infection by viruses.